# A positive feedback loop linking enhanced mGluR function and basal calcium in spinocerebellar ataxia type 2

Pratap Meera[1], Stefan Pulst[2], Thomas Otis[1,3]*

[1]Department of Neurobiology, Geffen School of Medicine, University of California, Los Angeles, United States; [2]Department of Neurology, University of Utah, Salt Lake, United States; [3]Neuroscience, Ophthalmology, and Rare Diseases, Roche Pharmaceutical Research and Early Development, Basel, Switzerland

**Abstract** Metabotropic glutamate receptor 1 (mGluR1) function in Purkinje neurons (PNs) is essential for cerebellar development and for motor learning and altered mGluR1 signaling causes ataxia. Downstream of mGluR1, dysregulation of calcium homeostasis has been hypothesized as a key pathological event in genetic forms of ataxia but the underlying mechanisms remain unclear. We find in a spinocerebellar ataxia type 2 (SCA2) mouse model that calcium homeostasis in PNs is disturbed across a broad range of physiological conditions. At parallel fiber synapses, mGluR1-mediated excitatory postsynaptic currents (EPSCs) and associated calcium transients are increased and prolonged in SCA2 PNs. In SCA2 PNs, enhanced mGluR1 function is prevented by buffering $[Ca^{2+}]$ at normal resting levels while in wildtype PNs mGluR1 EPSCs are enhanced by elevated $[Ca^{2+}]$. These findings demonstrate a deleterious positive feedback loop involving elevated intracellular calcium and enhanced mGluR1 function, a mechanism likely to contribute to PN dysfunction and loss in SCA2.

**\*For correspondence:**
thomas_stephen.otis@roche.com

**Competing interests:** The authors declare that no competing interests exist.

## Introduction

SCA2 is an autosomal dominant form of ataxia caused by expanded CAG-triplet repeats in the ataxin2 gene which are translated into abnormally long polyglutamine stretches in the ataxin2 protein, leading to a toxic gain-of-function. Mouse models of SCA2 exhibit age-dependent deterioration of motor function accompanied by transcriptional and electrophysiological changes that precede PN loss (*Kasumu and Bezprozvanny, 2012*; *Hansen et al., 2013*; *Dansithong et al., 2015*). These concerted cellular and physiological changes preceding the first signs of ataxia suggest that the disease is not driven solely by PN loss but may also be the result of PN dysfunction at the circuit level (*Meera et al., 2016*).

There is compelling evidence indicating that specific alterations in PN calcium homeostasis accompany pathogenesis of SCA2 (*Liu et al., 2009*; *Kasumu et al., 2012a*; *Dansithong et al., 2015*; *Halbach et al., 2017*). Furthermore, cerebellar transcriptome analyses have revealed associations between SCA genes and many genes involved in calcium homeostasis (*Lin et al., 2000*; *Serra et al., 2004*; *Hansen et al., 2013*; *Notartomaso et al., 2013*; *Bettencourt et al., 2014*; *Dansithong et al., 2015*; *Ingram et al., 2016*). Metabotropic type1 glutamate (mGluR1) receptors are highly expressed in PNs (*Masu et al., 1991*; *Shigemoto et al., 1992*) and deletion of the mGluR1 gene causes an ataxic phenotype (*Conquet et al., 1994*), while reintroduction of mGluR1 specifically in PNs restores motor coordination showing that mGluR1 in PNs is a key signaling molecule needed for normal development and function of the cerebellum (*Ichise et al., 2000*; *Nakao et al., 2007*).

In PNs, mGluR1 receptors are expressed both at parallel fiber (PF) and climbing fiber (CF) synapses (*Nusser et al., 1994*). At both types of excitatory synapse, glutamate release accesses AMPA receptors located immediately at postsynaptic density, and if released in amounts sufficient to overcome glutamate reuptake and clearance, mGluR1 receptors in the perisynaptic zone (*Brasnjo and Otis, 2001*; *Dzubay and Otis, 2002*). Activation of mGluR1 gives rise to a slow excitatory postsynaptic current mediated by TRPC3 channels (*Batchelor et al., 1994*; *Batchelor and Garthwaite, 1997*; *Hartmann et al., 2008*) and a local rise in calcium released from inositol-triphosphate receptor type 1 (IP$_3$R)-gated endoplasmic reticulum (ER) stores (*Batchelor et al., 1994*; *Finch and Augustine, 1998*; *Takechi et al., 1998*; *Hartmann et al., 2008*, Hartmann et al., 2011Hartmann et al., 2011*Hartmann et al., 2011*). Disruption of genes in the mGluR1 to IP$_3$R cascade results in cerebellar ataxia indication that this signaling pathway is pivotal for normal PN function and motor coordination (*Aiba et al., 1994*; *Conquet et al., 1994*; *Matsumoto et al., 1996*; *Offermanns et al., 1997*; *Kano et al., 1998*; *Ichise et al., 2000*; *Hartmann et al., 2004*; *Nakao et al., 2007*; *Hartmann et al., 2008*; *Becker et al., 2009*; *Sekerková et al., 2013*). Altered mGluR activity has been reported in SCA1 and SCA2 mouse models (*Liu et al., 2009*; *Power et al., 2016b*), while in humans, several rare forms of cerebellar ataxia result from mutations in genes in this cascade (*Yabe et al., 2003*; *van de Leemput et al., 2007*; *Guergueltcheva et al., 2012*; *Zanni et al., 2012*) implying that dysfunction in mGluR1 signaling and downstream calcium homeostasis could be common pathophysiological mechanisms in spinocerebellar ataxia (*Schorge et al., 2010*).

An unusual feature of mGluR1-IP$_3$R signaling that may be important for the chronic pathophysiology characteristic of SCAs is the robust positive feedback that links the G-protein and second messenger cascades with basal calcium levels. At an upstream point in the cascade, slight elevations of calcium strongly facilitate the TRPC3-mediated slow cation conductance (*Batchelor and Garthwaite, 1997*; *Dzubay and Otis, 2002*). Farther downstream, modestly elevated calcium levels enhance IP$_3$R function leading to larger calcium release from endoplasmic reticulum stores (*Bezprozvanny et al., 1991*; *Finch et al., 1991*; *Wang et al., 2000*; *Sarkisov and Wang, 2008*). Together these two positive feedback nodes have the potential to exacerbate any challenge to normal calcium homeostasis that might occur even as an indirect consequence of a chronic neurodegenerative condition (*Meera et al., 2016*).

Here we report that mGluR1 signaling is greatly amplified early in disease in a well-characterized transgenic mouse model of SCA2 expressing a human form of SCA2 with 127 polyglutamine repeats (*Hansen et al., 2013*). This is based on four observations: First, we show that the mGluR1 agonist dihydroxyphenylglycine (DHPG) evokes progressively larger increases in firing frequency in SCA2 versus wild type PNs across disease time course. Second, we find that across a range of firing frequencies, basal calcium concentrations are increased in SCA2 PNs. Third, PF-evoked mGluR1-mediated EPSPs and calcium transients are enhanced and prolonged in PNs of SCA2 mice. Finally, WT mGluR1 EPSCs are larger when calcium is buffered at elevated levels and the enhanced EPSCs in SCA2 PNs are normalized when calcium is strongly buffered at physiological levels. These results suggest that mGluR1 signaling in SCA2 PNs is hyperactive and there is a deleterious positive feedback loop involving calcium and mGluR1 activity. Our findings identify a plausible pathophysiological mechanism that could account for the vulnerability of PNs and the chronic and progressive nature of the disease.

## Results

### Activation of mGluR1 more strongly excites SCA2 PNs

Complex transcriptional changes involving genes related to the mGluR1 signal cascade have been reported in several SCA mouse models (*Lin et al., 2000*; *Serra et al., 2004*; *Hansen et al., 2013*; *Notartomaso et al., 2013*; *Bettencourt et al., 2014*; *Dansithong et al., 2015*); however, few studies have examined the physiological consequences of such changes. In order to determine the functional effects of changes in mGluR1 signaling, we designed a straightforward experiment to measure the effects on PN excitability of mGluR1 activation across age in a well-characterized SCA2 mouse model (*Hansen et al., 2013*).

We chose to monitor spontaneous action potential (AP) firing rate across age as this represents a minimally invasive and stable index of the excitability of individual PNs (*Häusser and Clark, 1997*;

*Raman and Bean, 1997*; *Smith and Otis, 2003*). Using extracellular recordings, PN cell firing was measured in acute cerebellar slices prepared either from mice expressing in PNs a SCA2 transgene with 127 polyglutamine encoding repeats (hereafter referred to as SCA2 mice) or in WT littermate control mice (*Hansen et al., 2013*). Slices were prepared from 8, 12, and 24 week old mice, an age range in this model that corresponds with steadily increasing transcriptional and physiological changes in PNs preceding the onset of marked behavioral ataxia (*Hansen et al., 2013*). At the youngest ages tested, application of 1 µM DHPG significantly increased PN firing frequency in SCA2 when compared to control littermates (*Figure 1*). The increased PN firing upon mGluR1 activation progressively increased in SCA2 slices from older animals, while PNs from WT littermates showed minimal change at the same ages (two way ANOVA indicates a significant effect of genotype p<0.0001, with posthoc Sidak's test indicating significance at specific ages; *Figure 1B*). These findings demonstrate that in this SCA2 model, increased responsiveness of SCA2 PNs to mGluR1 activation mirrors the progressive appearance of biochemical, physiological, and behavioral signs of pathology (*Hansen et al., 2013*).

## Somatic calcium concentrations are elevated across a range of firing frequencies in SCA2 PNs

We next set out to explore the interplay between action potential (AP) firing and calcium levels in normal and SCA2 PNs. In order to evaluate intracellular calcium levels over a physiological range of firing frequencies, we relied on whole cell current clamp recordings so that a range of steady state firing frequencies could be imposed on each neuron creating a frequency response curve for each PN. Whole cell recordings also allowed us to introduce the calcium indicator dye Oregon Green

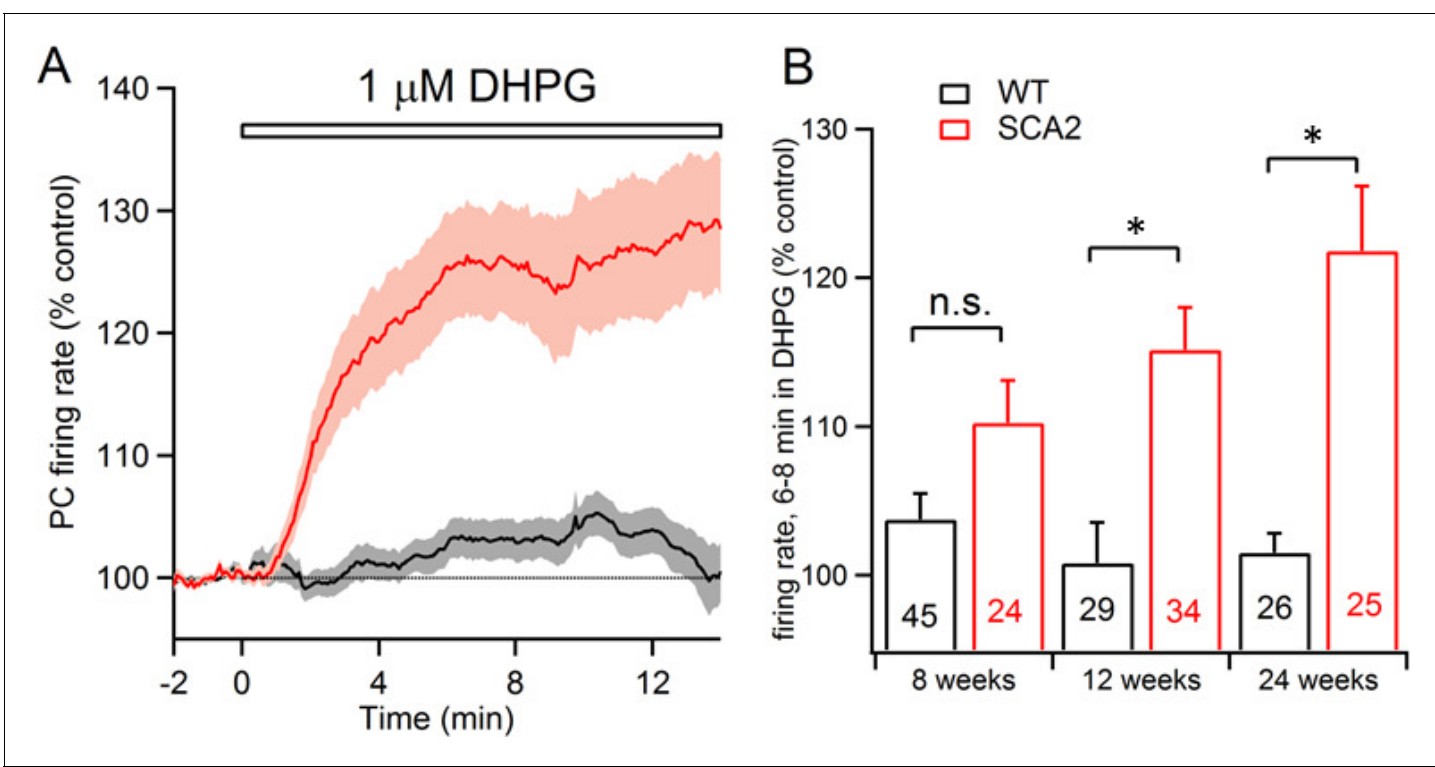

**Figure 1.** mGluR1-mediated excitation of PCs is enhanced in SCA2 mice. (**A**) Extracellular recordings of PN firing from cerebellar slices shows that application of a low concentration of the group1 mGluR1-selective agonist DHPG (1 µM) minimally excites control PNs (black line, grey indicates s.e.m.) but strongly excites PNs from SCA2 mice (red line, lighter red indicates s.e.m.). Data are mean AP firing rates normalized to pre-DHPG control period from 26 and 25 PNs in slices prepared from 24 week-old SCA2 and wild type mice. (**B**) Summary of mean normalized firing rate 6–8 min after application of 1 µM DHPG as a function of genotype (black=WT; red=SCA2) and age. Note the progressive increase in sensitivity to the mGluR1 agonist with age. Numbers of PNs recorded are listed on each bar; significance is indicated following a two way ANOVA test indicating a significant effect of genotype (p<0.0001) and Sidak's multiple comparison test (n.s. indicates non-significant; * indicates p<0.0002).

BAPTA-1 (OGB-1, 200 µM) and the red fluorescent dye Alexa 594 (20 µM) into each cell through the whole-cell pipette and use 2P microscopy to measure corresponding steady state calcium levels in the somatic compartment.

By injecting current through the whole-cell pipette we varied the steady state AP firing rate of each PN between 0 Hz, with resting membrane potential values between −65 and −75, and the maximal steady state firing rate, typically between 120 and 180 Hz. *Figure 2A–D* shows an example of such an experiment conducted on an eight week-old SCA2 PN. Displayed are the image of the PN (*Figure 2A*), the simultaneously measured firing rates (red) and relative OGB-1 fluorescence (green) over the time course of the experiment (*Figure 2B*), example traces showing steady state AP firing at the time points indicated by lowercase letters (*Figure 2C*). *Figure 2B* shows that rapid changes of firing rates (*Figure 2B*, red trace) induced by injection of hyperpolarizing current to silence the PN firing (e.g. the epoch between a and b), or injection of depolarizing currents to increase firing rates (e.g. epochs c and d) led to delayed changes in volume averaged calcium

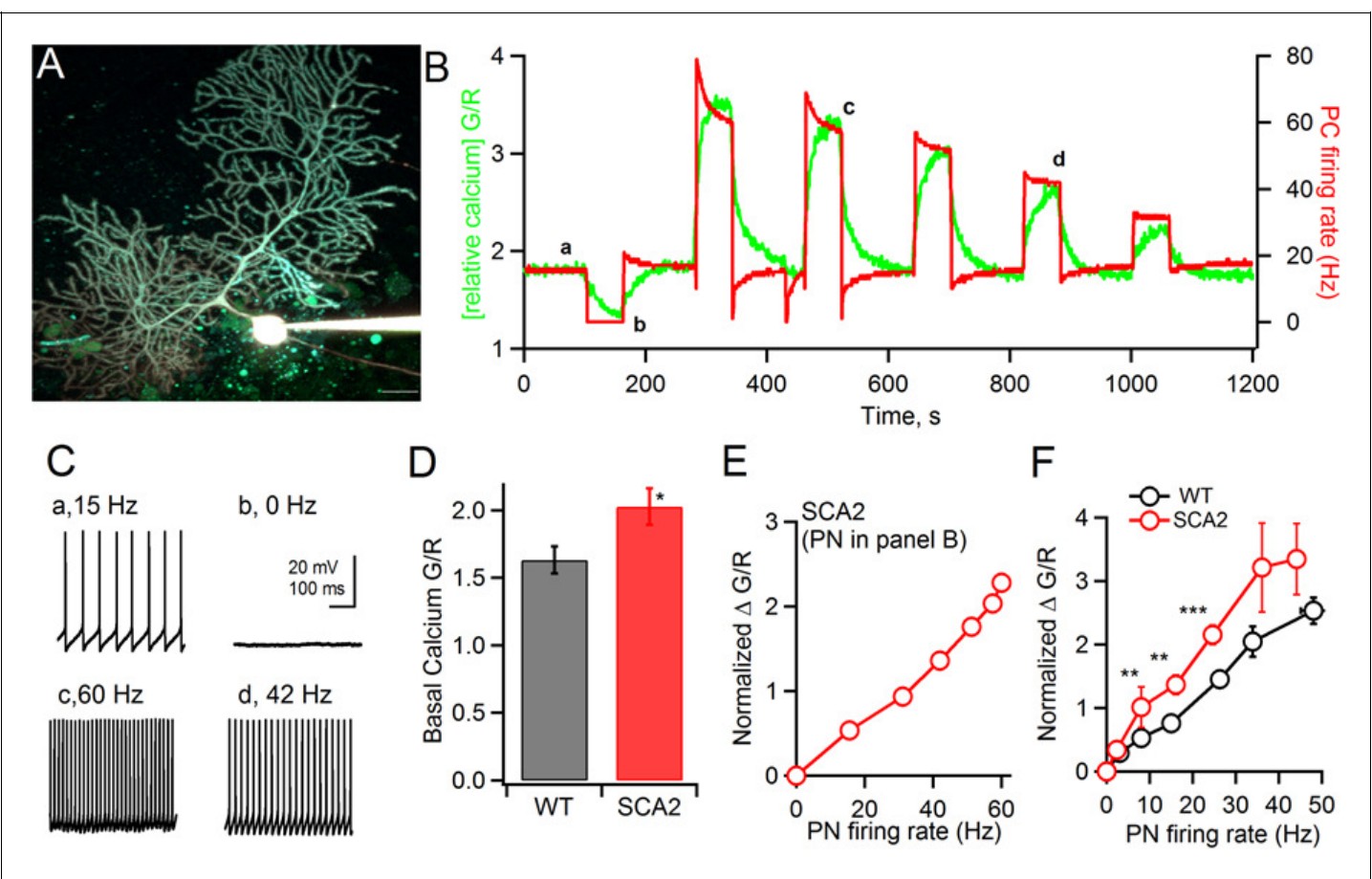

**Figure 2.** Two-photon calcium measurements show a steeper relationship between action potential (AP) firing frequency and steady-state calcium concentration in SCA2 PNs: (A) An 8 wk old SCA2 PN filled via whole-cell patch pipette with the calcium indicator dye OGB1 (200 µM) and the calcium–independent, red dye Alexa 594 (20 µM). Scale bar is 25 µm. (B) In green, changes in OGB1 fluorescence intensity normalized to Alexa 594 intensity (G/R) and, in red, the AP firing frequency in response to different current injections which impose a range of steady state AP firing rates. (C) Examples of firing rate at the indicated time points in panel B (a-d). (D) Quantification of mean basal calcium levels (G/R) at 0 Hz firing rate for WT (black 1.63 ± 0.098, n = 15) and SCA2 (red, 2.03 ± 0.13, n = 18). (E) The change in OGB-1 fluorescence relative to basal values at AP rate = 0 Hz (Normalized DG/R), for each firing rate measured in the PN in panel B. (F) Mean values of normalized ΔG/R versus AP firing rate for PNs from 8 to 10 week old mice, WT (black, mice = 15: PNs = 10 at 0 to 5 Hz, PNs = 8 at 6 to 10 Hz, PNs = 6 at 11 to 20 Hz, PNs = 12 at 21 to 30 Hz, PNs = 7 at 31 to 40 Hz, PNs = 2 at 40 to 50) and SCA2 (red, mice = 18, PNs = 28 at 0 to 5 Hz, PNs = 15 at 6 to 10 Hz, PNs = 17 at 11 to 20 Hz, PNs = 29 at 21 to 30 Hz, PNs = 5 at 31 to 40 Hz, PNs = 6 at 40 to 50 Hz). Significance levels indicated by * < 0.05, ** < 0.001, *** < 0.0001.

concentration (*Figure 2B*, green trace). The curve relating change in somatic calcium (normalized △G/R) as a function of firing rate for this PN is displayed in *Figure 2E*.

From these data we also evaluated whether baseline calcium levels differed between WT and SCA2 PNs when they were silenced (i.e. at firing rate 0 Hz). To estimate calcium in this condition we compared the ratio of somatic fluorescence values for OGB-1 and Alexa 594 expressed as G/R. Significantly higher G/R values were found for SCA2 PNs as compared to WT (SCA2, 2.03 ± 0.13, n = 18; WT, 1.63 ± 0.1, n = 15; p<0.05; *Figure 2D*). Even after subtracting these slightly higher baseline levels for SCA2 PNs and considering normalized changes in calcium as above, we found for SCA2 PNs a steeper relationship between AP firing rate and normalized △G/R across a range of physiological firing frequencies (*Figure 2F*). These findings support the hypothesis that calcium homeostasis is dysregulated early in the SCA2 disease process.

## Synaptic activation of mGluRs leads to enhanced slow EPSPs and larger intracellular calcium transients in SCA2 PNs

Prior work in another SCA2 mouse model with 58 polyglutamine repeats concluded that SCA2 proteins containing polyglutamine repeat expansions bind to and enhance IP3R function, thereby exacerbating mGluR1-initiated signaling in cultured PNs (*Liu et al., 2009*). To test whether mGluR1 signaling in PNs is hyperactive in SCA2 mice under more physiological circumstances we combined cerebellar slice electrophysiology and two photon calcium imaging to measure synaptically-activated, mGluR1-linked membrane conductances and intracellular calcium transients at different stages of pathology. Synaptic activation of mGluR1 was achieved by stimulating PFs with brief 100 Hz trains in the presence of a cocktail of receptor antagonists to block AMPA, NMDA and GABA$_A$ receptors (*Brasnjo and Otis, 2001*; *Karakossian and Otis, 2004*). Activation of mGluR1 leads to two distinct signals, a slow excitatory postsynaptic potential and a local dendritic calcium signal due to calcium release from the ER (*Batchelor and Garthwaite, 1997*; *Finch and Augustine, 1998*; *Takechi et al., 1998*; *Wang et al., 2000*; *Hartmann et al., 2008*, Hartmann et al., 2011Hartmann et al., 2011*Hartmann et al., 2011*). We recorded these downstream responses to synaptic mGluR1 activation in slices from 12 to 16 week old WT or SCA2 mice under whole cell current clamp mode with simultaneous 2P measurements of OGB-1 fluorescence. In local dendritic regions, 100 Hz stimulus trains of different durations gave rise to monotonically increased local calcium rises (△G/G) that were blocked by CPCCOET, an antagonist of mGluR1 (*Figure 3B,D and F*). Across this range of stimulus strengths, markedly enhanced calcium signals were observed in SCA2 PNs compared to WT PNs (*Figure 3E and F*). Significantly larger calcium transients in SCA2 PNs as compared to WT were also observed at 20–24 weeks in response to five pulse, 100 Hz stimulus trains (*Figure 3E and F*). These differences between genotypes were observed despite similar stimulus intensities in WT and SCA2 experiments (12–16 weeks, WT, 10.1 ± 0.47 μA, n = 23; SCA2, 10.0 ± 0.78 μA n = 20; 20–24 weeks, WT, 17.6 ± 4.2 μA, n = 8; SCA2, 17.1 ± 3.4 μA, n = 7).

Consistent with a comprehensive effect on different limbs of the mGluR1 signaling pathway, mGluR1-mediated synaptic currents were also robustly enhanced in SCA2 PNs. *Figure 4A and B* shows example slow EPSCs in WT and SCA2 PNs in response to 5 and 10 pulse trains of 100 Hz stimuli delivered to PFs. Similar to the mGluR1-mediated calcium transients, SCA2 PNs exhibited significantly larger EPSCs as compared to WT PNs across stimulus strengths (*Figure 4B*). Notably, larger EPSCs are not simply a consequence of increased calcium mobilization as intracellular calcium increases are not required for activation of the TRPC3 channel (*Dzubay and Otis, 2002*; *Hartmann et al., 2008*, Hartmann et al., 2011Hartmann et al., 2011*Hartmann et al., 2011*). Thus, these results suggest a comprehensive increase in mGluR1-mediated signaling, possibly due to some positive feedback occurring early in the signal transduction cascade.

## Basal calcium levels determine the larger mGluR1-mediated responses in SCA2 PNs

Given these findings of dysregulated calcium and enhanced mGluR1-mediated synaptic responses in SCA2 PNs, and considering the positive feedback mechanisms linking basal calcium levels and mGluR function (*Bezprozvanny et al., 1991*; *Finch et al., 1991*; *Batchelor and Garthwaite, 1997*), we sought to test whether the two physiological abnormalities were causally related. We reasoned that if calcium levels were responsible for enhancing mGluR responses in SCA PNs, then buffering

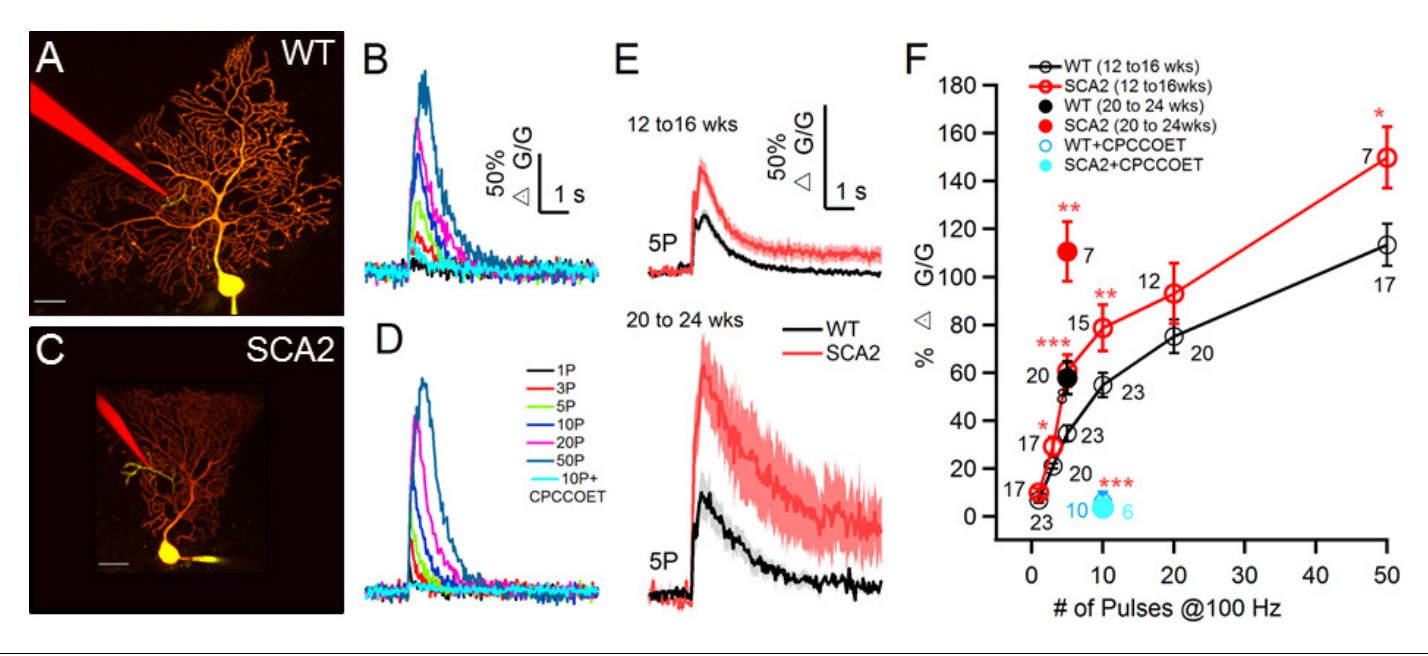

**Figure 3.** Synaptic mGluR1- mediated calcium transients are significantly increased and prolonged in SCA2 PNs. (A and C). Two photon images of 12 wk old WT (A) and SCA2 (C) PNs filled via patch pipette (yellow) with the calcium indicator dye 200 µM OGB1 and 20 µM Alexa 594. A second pipette (red) filled with Alexa 594 is placed in the dendritic region to minimally stimulate PF synaptic inputs. Note the local region of $[Ca^{2+}]$ rise near the tip of the stimulating electrode (green). Scale bars are 25 µm in both images. (B and D) Intracellular $Ca^{2+}$ signals ($\Delta G/G$) for responses from WT (B) and SCA2 (D) PNs elicited by stimulation of PFs with 100 Hz trains and indicated number of pulses in the presence of AMPA, NMDA, and $GABA_A$ receptor antagonists. Note, $Ca^{2+}$ signals are blocked by the mGluR1 antagonist CPCCOET (cyan traces). (E) Mean ± s.e.m. changes in $Ca^{2+}$ in response to PF stimulation (5 pulses at 100 Hz) across experiments for indicated age groups and genotypes. WT (black lines, grey indicates s.e.m.) and SCA2 (red lines, lighter red indicates s.e.m.). (F) Summary across a range of train lengths of PF-evoked $Ca^{2+}$ transients at age 12–16 weeks shows larger transients in SCA2 PNs. Open red symbols denote SCA2 and black symbols WT PNs. Larger $Ca^{2+}$ transients are also seen for SCA2 PNs at age 20–24 weeks in response to five pulse trains (solid red vs. black symbols). Responses are blocked by CPCCOET (WT, blue and SCA2, cyan) confirming mGluR1 dependence. Numbers of synaptic sites recorded are listed next to each symbol (total number of PNs are 21 from 13 WT mice and 12 PNs from 10 SCA2 mice at 12 to 16 wks old, 8 PNs for WT and seven for SCA2);* indicate significance *p<0.05, **p<0.01, ***p<0.001.

basal calcium concentration at a physiological level (~140 nM) might prevent the increase in mGluR function observed in SCA2 PNs and conversely that buffering calcium at elevated levels (~450 nM) in WT PNs might mimic the enhancement observed in SCA2 PNs.

Distinct pipette solutions were formulated and calibrated to confirm 'physiological' and 'elevated' calcium levels; in addition, we blocked glutamate uptake by addition of TBOA to the extracellular solution in order to rule out contributions of possible SCA2-related changes in glutamate transporters on mGluR1-mediated synaptic responses (see Materials and methods). Although it is not possible to make measurements of mGluR1-mediated calcium transients with this approach, we measured mGluR1 EPSCs in voltage clamp mode and found that when internal calcium is clamped with the 'physiological' pipette solution, the significant enhancement was abolished and SCA2-Q127 and WT mGluR EPSCs showed similar amplitudes (*Figure 5A and C*). Conversely, buffering intracellular calcium to elevated levels (~450 nM) increased the magnitude of mGluR1 EPSCs in WT PNs relative to those recorded with calcium buffered at physiological levels (*Figure 5B*). Taken together, these results indicate that an elevated intracellular calcium concentration contributes to the increased mGluR function observed in SCA2 PNs. The findings also support the hypothesis that positive feedback between basal calcium levels and mGluR1 signaling efficacy is a deleterious pathophysiological consequence of the disease which could drive PN dysfunction and may play a key role in the progressive loss of PNs that characterize all forms of SCA.

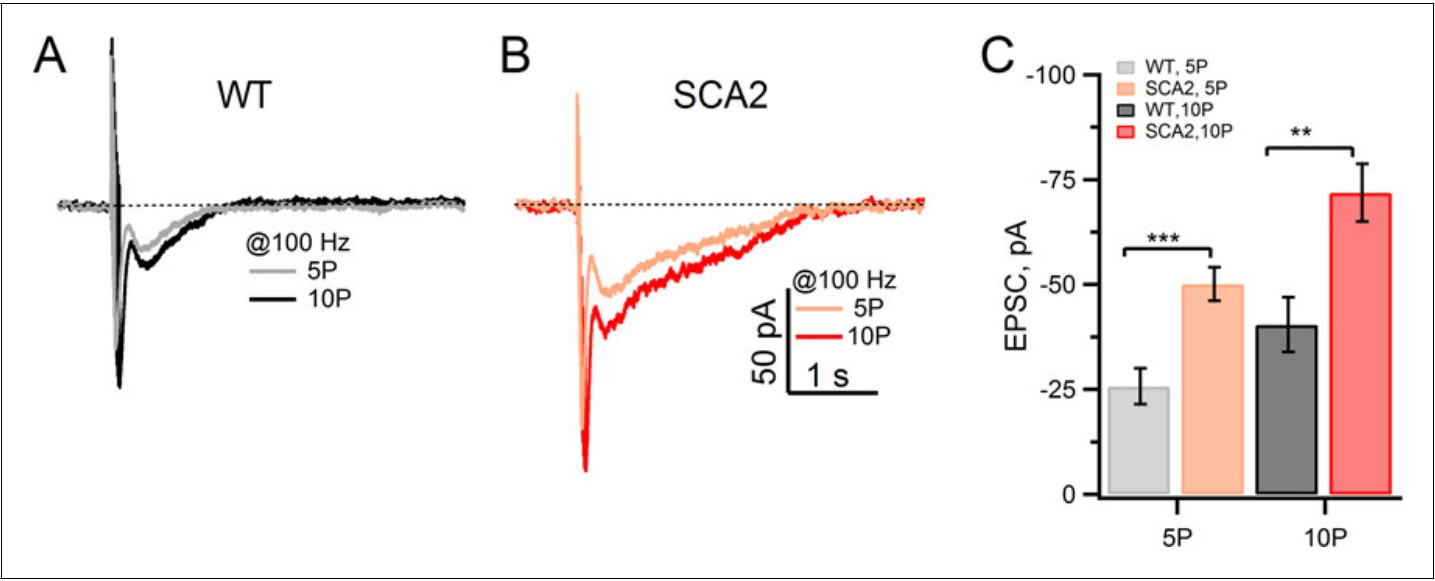

**Figure 4.** Synaptic mGluR1 EPSCs are significantly larger in SCA2 PNs. (**A** and **B**) Slow mGluR1 EPSCs in response to 5 and 10 pulse, 100 Hz stimuli applied to PFs. mGluR1 EPSCs are measured with 1 mM EGTA (internal pipette solution) and in extracellular solution containing antagonists for GABA$_A$ receptors (100 µM PTx), AMPA receptors (2 µM DNQX, leading to a partial block,) and glutamate transporters (100 µM DL-TBOA). (**C**) Average peak mGluR1 EPSCs in response to 5 or 10 pulses @100 Hz for WT and SCA2 recordings. Average peak values are −26 ± 4 pA, and −40 ± 7 pA in WT mice (n = 14) and −50 ± 4 pA and −72 ± 7 pA in SCA2-Q127 (n = 9) at 5 and 10 pulses, respectively. Significance levels are indicated by *p<0.05, **p<0.01, ***p<0.001.

## Discussion

### Enhanced mGluR signaling in SCA2

In the present study we present multiple lines of evidence that hyperactivity of the mGluR1/IP$_3$R pathway in PNs is an underlying consequence of SCA2 pathogenesis. In SCA2 mice, we observed an age-dependent increase in PN responsiveness to low concentrations of the mGluR1 agonist DHPG, an effect not seen for WT PNs (*Figure 1*). Examination of synaptically-elicited mGluR1-mediated responses further confirmed this hyperactivity in that both mGluR-linked calcium mobilization and mGluR-coupled TRPC3 membrane conductances were progressively and robustly enhanced in SCA2 PNs relative to PNs from WT littermates. The findings are consistent with the time course of transcriptional, biochemical, electrophysiological, and behavioral changes described in prior work on this mouse line; at four weeks transcriptional dysregulation and slowed spontaneous firing are first detectable, followed soon after by behavioral signs of ataxia, features that worsen in parallel over the next 20 weeks (*Hansen et al., 2013*; *Dansithong et al., 2015*).

These observations extend prior work demonstrating that ataxin2 proteins with expanded polyglutamine repeats augment IP$_3$R function and lead to enhanced mGluR-mediated calcium transients in cultured PNs (*Liu et al., 2009*); the finding are also consistent with subsequent work showing therapeutic effects in an SCA2 mouse model upon delivery to PNs of an enzyme to degrade IP$_3$ (*Kasumu et al., 2012a*). Finally, our findings nicely complement recent observations demonstrating enhancement of mGluR1 signaling in a conditional SCA1 model (*Power et al., 2016b*). Collectively these studies strengthen the hypothesis that dysregulation of mGluR signaling is a common, exacerbating factor in multiple forms of SCA (*Meera et al., 2016*; *Power et al., 2016a*), especially when considered in a larger context with monogenetic forms of ataxia which directly affect the mGluR1 and downstream genes (*Yabe et al., 2003*; *van de Leemput et al., 2007*; *Guergueltcheva et al., 2012*).

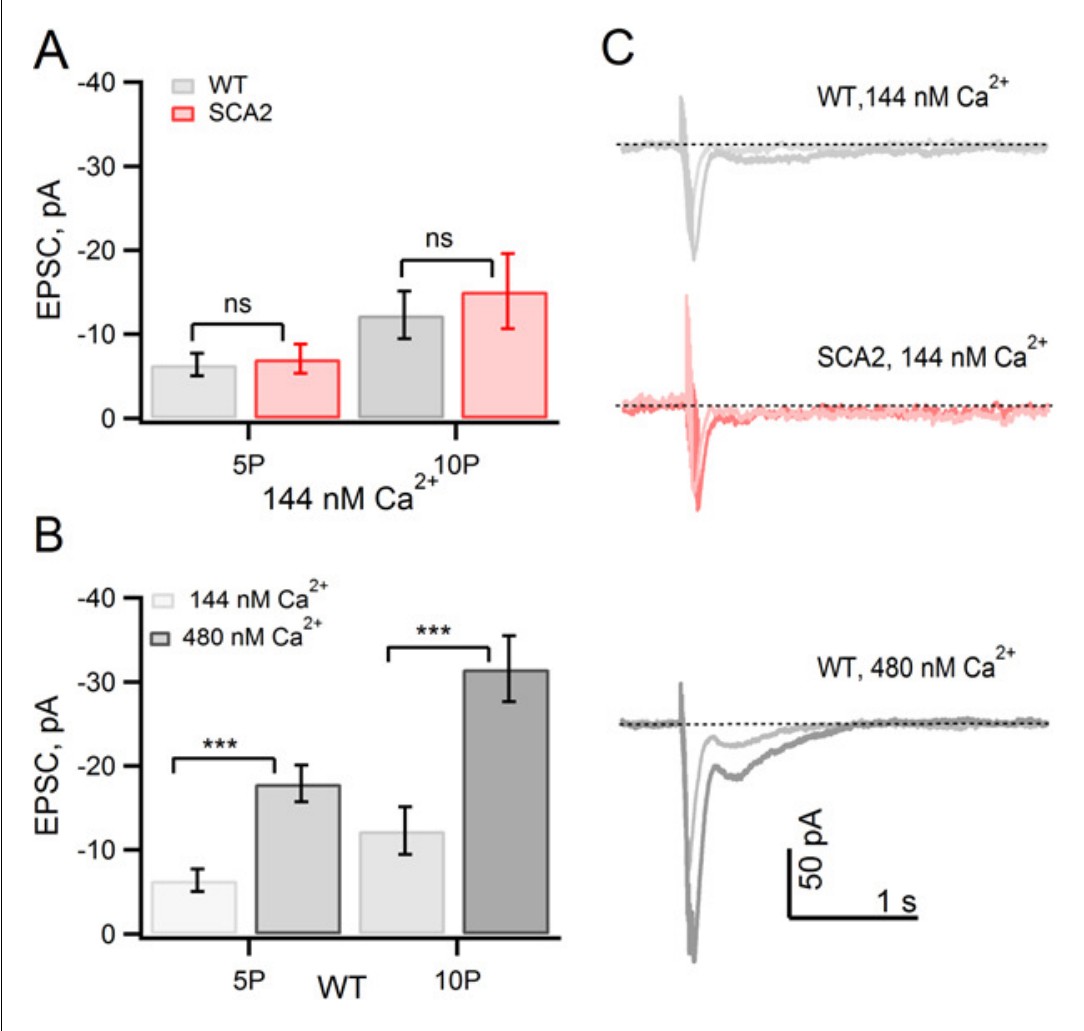

**Figure 5.** Elevated basal internal [Ca$^{2+}$] levels in SCA2 PNs account for enhanced mGluR1 EPSCs. (**A**) mGluR1 EPSCs are similar both in WT and SCA2 when internal free [Ca$^{2+}$] is strongly buffered with 10 mM internal BAPTA. Mean values of PF-evoked slow mGluR1 EPSCs in WT (5P, n = 23; 10P, n = 19) versus SCA2 (5P, n = 26; 10P, n = 16). (**B**) Buffering internal free [Ca$^{2+}$] to an elevated level of 480 nM in WT PNs increases peak mGluR1 EPSCs. Mean mGluR1 EPSCs at 5P is −18 ± 2 (n = 12 at 5P) and −32 ± 4 (n = 11 at 10P) pA. (**C**) Representative synaptic mGluR1 EPSCs from PNs recorded with free Ca$^{2+}$ strongly buffered at 144 nM (WT and SCA2) or 480 nM (WT) levels (see Materials and methods). These findings are consistent with the hypothesis that the observed increased mGluR1 EPSCs in SCA2 mice is due to an elevated internal [Ca$^{2+}$] in PNs; significance of p<0.001 indicated by ***.

## Dysregulated calcium homeostasis in SCA2

Our findings are fully consistent with emerging evidence showing that abnormalities in PN calcium signaling play a significant role in pathogenesis of human ataxias (*Yabe et al., 2003*; *Serra et al., 2004*; *van de Leemput et al., 2007*; *Carlson et al., 2009*; *Liu et al., 2009*; *Schorge et al., 2010*; *Guergueltcheva et al., 2012*; *Kasumu et al., 2012a*; *Zanni et al., 2012*; *Bettencourt et al., 2014*; *Fogel et al., 2015*). Mutations in several genes with known functions related to calcium homeostasis cause monogenetic forms of ataxia, including those encoding the P type voltage-gated Ca$^{2+}$ channel (*Zhuchenko et al., 1997*), mGluR1 (*Guergueltcheva et al., 2012*), the type 1 IP$_3$R (*van de Leemput et al., 2007*), the type 3 Ca$^{2+}$ ATPase (*Zanni et al., 2012*; *Calì et al., 2015*), and possibly TRPC3 (*Fogel et al., 2015*). Moreover, cerebellar PNs abundantly express many genes involved in calcium homeostasis and a large fraction of these genes are transcriptionally dysregulated in various SCAs (*Serra et al., 2004*; *Schorge et al., 2010*; *Bettencourt et al., 2014*; *Ingram et al., 2016*). Downregulation of the gene encoding the calcium binding protein Calbindin-28K is commonly observed in a

number of SCAs (*Vig et al., 1998*; *Serra et al., 2004*; *Hansen et al., 2013*; *Bettencourt et al., 2014*; *Ingram et al., 2016*).

In mice, spontaneously occurring and transgenic alteration of numerous genes related to calcium homeostasis lead to altered PN physiology, morphology, and to behavioral ataxia. For example, the *moonwalker* mouse model carries a mutation leading to constitutive activation of TRPC3 channels and a dominantly inherited cerebellar ataxia (*Becker et al., 2009*; *Becker, 2014*); perhaps relatedly, mutated forms of PKCγ found in SCA14 fail to phosphorylate TRP channels resulting in sustained $Ca^{2+}$ entry into PNs (*Adachi et al., 2008*). Spontaneous mutations in IP$_3$R1 cause ataxia in mice (*Street et al., 1997*). Deletion of STIM1, a protein required to refill ER stores (*Hartmann et al., 2014*), or deletion of PMCA2, a plasma membrane $Ca^{2+}$ ATPase pump (*Empson et al., 2010*) lead to ataxic phenotypes. Finally, deletion of calcium buffering proteins like calbindin-28K or parvalbumin cause ataxia (*Schwaller et al., 2002*) and deletion of one copy of calbindin-28K accelerates the ataxia phenotype in SCA1 mice (*Vig et al., 1998*, *2001*, *2012*). In conjunction with this converging evidence, the results reported here strengthen the case that disturbed neuronal $Ca^{2+}$-signaling plays an important role in degenerative conditions involving PNs, particularly in the pathology of several forms of SCA.

## A deleterious positive feedback loop between calcium and mGluR1 signaling

Our evidence suggests that SCA2 is likely exacerbated by prominent positive feedback mechanisms exerted by elevated basal calcium on mGluR1 coupling to TRPC3 channels (*Batchelor and Garthwaite, 1997*) and to IP$_3$R-mediated release of intracellular calcium (*Bezprozvanny et al., 1991*; *Finch et al., 1991*; *Wang et al., 2000*; *Sarkisov and Wang, 2008*). We found that the enhancement of mGluR EPSCs in SCA2 relative to WT PNs was abolished when calcium was buffered to normal resting levels and in WT PNs, mGluR EPSCs could be enhanced by elevating basal calcium. These results build on prior reports showing that transient elevations of calcium can enhance coupling to the TRPC3 channels responsible for the slow EPSP (*Batchelor and Garthwaite, 1997*) and can enhance IP$_3$ actions at the IP$_3$R (*Wang et al., 2000*; *Sarkisov and Wang, 2008*). They are also consistent with observations showing that the two downstream limbs of the mGluR1 signaling cascade, to TRPC3 channels and to PLCβ/IP$_3$R signaling, can operate independently of one another such that elimination of TRPC3 does not affect IP$_3$R-mediated calcium release and strongly buffering calcium at permissive levels can support robust slow EPSCs (*Dzubay and Otis, 2002*; *Hartmann et al., 2008*, Hartmann et al., 2011Hartmann et al., 2011*Hartmann et al., 2011*).

Considering the similarities between our findings and those recently reported for an SCA1 model which showed prolonged mGluR1-mediated PF slow EPSCs and calcium transients and which demonstrated improved motor function when treated with an mGluR1 antagonist (*Power et al., 2016b*), it is tempting to generalize. These convergent sets of findings suggest a model in which mechanistically diverse insults to PNs, not only those limited to direct challenges to calcium homeostasis pathways, could lead to cellular stress, modest elevations in basal calcium concentrations, and through the mechanisms presented here, hyperactive mGluR1 signaling cascades. Enhanced mGluR1 signaling to TRPC3 channels and IP$_3$Rs would in turn exacerbate the calcium dysregulation. We hypothesize that such a positive feedback loop may contribute to the progressive course of SCAs because over time individual PNs would become more impaired, and more PNs would be affected.

The effects of hyperactive mGluR signaling on PN excitability are more complex. PNs have a variety of interdependent ionic conductances mediated by an array of voltage-gated and calcium-activated ion channels. For example, inhibition or activation of the prominent SK-type calcium activated $K^+$ conductance leads to robust changes in PN firing frequency (*Womack and Khodakhah, 2003*; *Walter et al., 2006*; *Kasumu et al., 2012b*; *Maiz et al., 2012*; *Egorova et al., 2016*). We hypothesize that, driven by elevated basal calcium levels, SK type $K^+$ conductances are tonically activated and that the resulting membrane hyperpolarization contributes to the slowing of spontaneous PN firing observed in slices from SCA2 mice (*Meera et al., 2016*) as well as to the aberrant firing patterns observed with intact circuits in vivo (*Egorova et al., 2016*). The resulting progressive loss of inhibitory output from the cerebellar cortex is predicted to degrade the capability of the cerebellum to contribute to motor behavior thus leading to ataxia.

# Materials and methods

### SCA2 mouse line

The SCA2 mouse model (RRID:MGI:5467323) was generated with standard transgenic mouse approaches (*Hansen et al., 2013*). In addition to mouse ataxin2, PNs from this SCA2 mouse model express the complete human ataxin-2 cDNA encoding 127 glutamine repeats (Q127) under the control of the PN-specific Pcp2 promoter.

### Preparation of cerebellar slices

Acute parasagittal slices of 285 µm thickness were prepared from the cerebellum of SCA2 mice and control littermates by following published methods (*Hansen et al., 2013*). In brief, cerebella were quickly removed and immersed in an ice-cold extracellular solution with composition of (mM): 119 NaCl, 26 NaHCO$_3$, 11 glucose, 2.5 KCl, 2.5 CaCl$_2$, 1.3 MgCl$_2$ and 1 NaH$_2$PO$_4$, pH 7.4 when gassed with 5% CO$_2$/95% O$_2$). Cerebella are sectioned using a vibratome (Leica VT-1000, Leica Biosystems, Nussloch, Germany). Slices were initially incubated at 35°C for 30 min, and then stored at room temperature until use.

### Extracellular electrophysiology

Extracellular recordings were obtained from PNs in slices constantly perfused with carbogen-bubbled extracellular solution as above with additional compounds as indicated. Cells were visualized with DIC optics and a water-immersion 40× objective (NA 0.75). Glass pipettes of ~3 MΩ resistance (Model P-1000, Sutter instruments, Novato, CA) when filled with extracellular solution were positioned near PN axon hillocks in order to measure action potential-associated capacitive current transients in voltage clamp mode with the pipette potential held at 0 mV. Data was acquired using a Multiclamp 700B amplifier at 20 kHz, Digidata 1440 with pClamp10 (Molecular Devices, Sunnyvale, CA) and filtered at 4 kHz. A total of 20 to 45 PNs were measured from each genotype and each recording was for 10 to 15 min. For each genotype 6 to 8 mice were used and the experimenter was blinded to the genotype.

### Whole cell electrophysiology and calcium imaging

Using a two-photon microscope and standard electrophysiology set-up, calcium levels were measured from somatic regions for experiments monitoring the relationship between spiking and calcium (*Figure 2*) and from dendritic regions in experiments measuring PF synaptic calcium transients (*Figure 3*). In all experiments, PNs were held in either whole cell current clamp or voltage clamp mode under the indicated conditions. PNs were filled with a pipette solution consisting of (in mM): 135 KMS0$_4$, NaCl, 10 HEPES, 3 MgATP, 0.3 Na2GTP, 200 µM OGB1 and 20 µM Alexa 594. Pipette solution was allowed to equilibrate for at least 30 min in whole cell mode before calcium measurements were undertaken. Unless otherwise indicated, relative fluorescence is expressed as OGB1 intensity (G or △G) normalized either to baseline OGB1 intensity (G) or to Alexa 594 intensity (R). Ratios of fluorescence for OGB-1 alone (△G/G) or for bimolecular measurements (G/R) were collected at frame rates of 1/s using 810 nm illumination to excite both fluorescent dyes. Using OGB-1 as an indicator, such ratios can report steady-state calcium concentrations from normal resting values up to ~1 µM (*Maravall et al., 2000*).

In experiments designed to test the relationship between steady-state spiking rate and somatic calcium levels (*Figure 2*), spiking rate was varied by injecting fixed amounts of current ranging from −100 pA to +600 pA in 100 pA increments.

Simultaneous mGluR EPSPs and calcium were measured in presence of GABA$_A$ receptor antagonist, picrotoxin (PTX at 100 µM), AMPA receptor blockers (5 µM NBQX and 10 µM DNQX). In addition, in *Figure 5* the glutamate transporter blocker TBOA (100 µM) was included in extracellular solutions in order to rule out contributions of glutamate reuptake to mGluR EPSC amplitudes. PF synaptic responses were elicited by minimal stimulation (typically 5–25 µA for 100 µs) from a patch pipette placed in the dendritic region; pipettes were filled with extracellular solution containing 20 µM Alexa 594 in order to aid 2P visualization of the stimulating pipette. PF-mediated mGluR EPSPs/EPSCs were evoked by stimulation of PFs with 100 Hz trains, with the indicated numbers of pulses. Corresponding intracellular Ca$^{2+}$ signals (ΔG/G or R) for responses for WT and mutant mGluR EPSPs

were abolished by the mGluR1 antagonist CPCCOET. The composition of the pipette solution with 10 mM BAPTA for measuring mGluR EPSCs was (mM): 95 KMSO4, 10 BAPTA tetrapotassium salt, 10 NaCl, 20 HEPES, 3 MgATP, 0.3 Na2GTP, pH7.3. $CaCl_2$ was adjusted to the desired pCa by calculating according to WebMAXC Standard (http://web.stanford.edu/~cpatton/webmaxcS.htm) and free calcium was checked with a calcium electrode (World Precision Instruments, USA). Experiments were analyzed using Clampfit (Molecular Device) and IgorPro (Wavemetrics) and further analyzed using Microsoft Excel; figures were made in IgorPro. Calcium signals were analyzed using Slidebook (Intelligent Imaging Innovations, Inc., Boulder, CO). Results are presented as mean ±SEM. Chemicals purchased were from Sigma Aldrich, Tocris, and Invitrogen, USA.

## Additional information

### Funding

| Funder | Grant reference number | Author |
|---|---|---|
| NIH Office of the Director | NS 033123 | Stefan Pulst Thomas Otis |
| NIH Office of the Director | NS 090930 | Thomas Otis |

The funders had no role in study design, data collection and interpretation, or the decision to submit the work for publication.

### Author contributions

PM, Formal analysis, Investigation, Writing—review and editing; SP, Conceptualization, Funding acquisition, Writing—review and editing; TO, Conceptualization, Formal analysis, Supervision, Funding acquisition, Writing—original draft, Project administration, Writing—review and editing

### Author ORCIDs

Thomas Otis, http://orcid.org/0000-0003-0383-8928

### Ethics

Animal experimentation: This study was performed in strict accordance with the recommendations in the Guide for the Care and Use of Laboratory Animals of the National Institutes of Health. All of the animals were handled according to approved institutional animal care and use committee (IACUC) protocols (#08-133) of the University of California Los Angeles. The protocol was approved by the Chancellor's Animal Research Committee (Permit Number: 1998-139).

## Additional files

### Supplementary files

• Supplementary file 1. Statistics for *Figures 1–5* and power analysis for the data presented in *Figure 1*.

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
