## [Decision Letter]

Thank you for submitting your article "A positive feedback loop linking enhanced mGluR function and basal calcium in spinocerebellar ataxia type 2" for consideration by *eLife*. Your article has been reviewed by two peer reviewers, and the evaluation has been overseen by a Reviewing Editor and Gary Westbrook as the Senior Editor. The following individuals involved in review of your submission have agreed to reveal their identity: Elizabeth Finch (Reviewer #1) and Ruth Empson (Reviewer #2).

The reviewers have discussed the reviews with one another and the Reviewing Editor drafted this decision to help you prepare a revised submission.

Summary:

The manuscript tests the hypothesis that mGluR1-mediated calcium signaling is dysregulated in cerebellar Purkinje neurons (PNs) in an established mouse model of SCA2. To address this question, the authors use a combination of electrophysiological recording approaches and 2-photon microscopic calcium imaging in acute cerebellar slices. They demonstrate that mGluR-mediated calcium signaling is enhanced under various physiological conditions and that this dysregulation of calcium homeostasis increases with age. The authors propose that this dysregulation of mGluR-mediated calcium signaling involves positive feedback mechanisms based on calcium-dependent facilitation of the TRPC3-mediated cation conductance and of IP3 receptor activity and that this positive feedback loop is likely to contribute to progressive PN dysfunction and eventual loss in SCA2.

Essential revisions:

The reviewers were favorable about the quality and the import of the study. However, they requested that a few points be clarified or discussed further, which are summarized here and expanded upon in their own words below.

1) Please comment on the reduced calbindin expression in this SCA2 mouse, which may exacerbate the proposed feedback cycle.

2) Provide more details of experiments of Figure 3, regarding normalizing measurements between cells and genotypes, range of stimulus intensities, input-output analyses between genotypes (if performed).

3) Please comment on the observed reduction in Purkinje neuron phasic firing in SCA2 as early as 6-8 weeks, in contrast to the results here, which would predict enhanced firing frequency, ideally integrating with existing literature.

Combined general assessment:

The manuscript by Meera et al. tests the hypothesis that mGluR1-mediated calcium signaling is dysregulated in cerebellar Purkinje neurons (PNs) in an established mouse model of SCA2. To address this question, the authors use a combination of electrophysiological recording approaches and 2-photon microscopic calcium imaging in acute cerebellar slices. The authors demonstrate that mGluR-mediated calcium signaling is enhanced under various physiological conditions and that this dysregulation of calcium homeostasis increases with age. Specifically, they show that in SCA2 compared to WT PNs 1) the mGluR1 agonist DHPG produced a larger enhancement of AP firing rate and this enhancement increased with age; 2) somatodendritic calcium levels are increased in both the absence and presence of APs; 3) PF-evoked mGluR-mediated EPSPs and calcium transients are increased. 4) They further show that buffering calcium to elevated levels increases EPSC amplitudes in WT PNs and, conversely, strong buffering of calcium to physiological levels normalizes EPSC amplitudes in SCA2 PNs. The authors propose that this dysregulation of mGluR-mediated calcium signaling involves positive feedback mechanisms based on calcium-dependent facilitation of the TRPC3-mediated cation conductance and of IP3 receptor activity and that this positive feedback loop is likely to contribute to progressive PN dysfunction and eventual loss in SCA2.

This study make a significant contribution to our understanding of the physiological mechanisms that contribute to SCA2 and is a valuable addition to the increasing body of evidence that impaired calcium homeostasis is likely to be a common pathogenic mechanism that contributes to many forms of ataxia. Overall, the experiments are well-designed to test key tenets of the hypothesis, the major conclusions are well-supported by presented data, the figures are clear and informative, and the manuscript tells a clear and logical story. The authors also provide an informative discussion of the contribution of this study to outstanding questions in the field and present an intriguing model for the role of PN calcium dysregulation in the pathogenesis of ataxias. This reviewer has no major concerns with this study or the manuscript.

This manuscript shows enhanced metabotropic glutamate signalling in SCA2, which is sufficient to drive mGluR1 mediated Purkinje neuron firing output. The authors then go on to show that the source of this enhanced excitability in SCA2 is a larger and prolonged mGluR1 synaptic current. Then with carefully controlled calcium titration experiments using 2P imaging the authors elegantly show that the underlying cause of the enhanced current is elevated cytosolic calcium. Indeed they nicely show that buffering calcium high to 450 nM in a wild type Purkinje neuron enhances the mGluR1 current to a similar manner as in SCA2, and conversely that buffering calcium low to 144 nM normalises the enhanced mGluR1 current in the SCA2 Purkinje neurons. The authors make the argument that it is raised calcium that is causing this positive feedback loop, enabling a larger inward current and greater firing in a vicious cycle that ultimately leads to the demise of the Purkinje neuron.

I am very enthusiastic that this work gets published in a high profile journal. Within the literature and the mGluR1/ataxia community there is considerable controversy that needs further discussion; the reason for the controversy lies in the established / traditional idea that mGluR1 signalling is lost in many forms of ataxia, most notably that the mGluR1 knockout mouse is ataxic. Importantly, and against this tradition, here the authors show that mGluR1 signalling is ENHANCED early in the disease, and that this elevation is critical for disease progression, EVEN in the face of reduced mGLuR1 expression in this (and other) models – I note that grm1 mRNA expression (mGluR1 mRNA) is reduced in SCA2 (Hansen et al). It is important within the field that this critical distinction of mGluR1 changes in early versus late stages of ataxia is recognized, and in this study enhanced activity driving disease progression is elegantly made. In addition, the concept that pathophysiological activity can be enhanced even if expression is reduced also needs to be made and more widely accepted/understood. Critically, the idea that mGluR1 signalling activity is enhanced in the early stages of ataxia is in agreement with several more recent clinical and pre-clinical studies pointing to a similar phenomenon, including our own work in SCA1.

The paper is also significant because it identifies over active mGluR1 signalling as a common mechanism amongst several pre-clinical forms of ataxia (including SCA1) and so could represent an exciting target for the early remedy of many ataxias. This is particularly relevant since although ataxias are rare, exploring viable common and "druggable" mechanisms is a priority. The particularly high economic and patient care burden of rare diseases such as ataxia is becoming increasingly recognized.

I have no substantive concerns about the quality of the work, the work is beautifully carried out and the results and arguments are persuasive.

Detailed points for essential revisions:

1) What the authors cannot conclude is the source of the raised calcium, but they make the inference that it is from IP3R dependent release from internal stores, based upon the literature and known signalling changes in SCA2. They speculate about this in the Discussion, and most likely there are multiple sources. Perhaps the authors should comment on the reduced calbindin expression in this SCA2 mouse, which would certainly exacerbate the feedback cycle that the authors propose.

2) In the experiments shown in Figure 3, more details are needed about how these experiments were done and how they were normalized between cells and genotypes. What was the range of stimulus intensities used for PF stimulation? Did the authors perform an input-output analysis? If so, was there a difference between genotypes? How were calcium signals normalized between cells/genotypes?

3) The authors should comment on the observed reduction in Purkinje neuron phasic firing in SCA2 as early as 6-8 weeks, whereas in contrast the results here would predict enhanced firing frequency. Could the enhanced mGluR1 currents, and therefore Purkinje neuron excitability be particularly important during behaviourally relevant changes in Purkinje neuron firing behaviour e.g. after high frequency parallel fibre input, in response to a sensory stimulation? Or is tonic mGluR1 activity enhanced and firing frequency depressed through an alternative mechanism? (E.g. increase KCa channel current?)

---

## [Author Response]

*Essential revisions:*

*The reviewers were favorable about the quality and the import of the study. However, they requested that a few points be clarified or discussed further, which are summarized here and expanded upon in their own words below.*

*1) Please comment on the reduced calbindin expression in this SCA2 mouse, which may exacerbate the proposed feedback cycle.*

This is an excellent point. Calbindin expression decreases in a number of models of SCA. Although some of the decrease is due to the shrinkage of the PN dendritic tree and loss of PNs, there is evidence to suggest that expression is reduced within individual PNs, which could impact basal ^Dis^Ca^2+^ levels in these cells. We have now added text to the Discussion section entitled “Dysregulated calcium homeostasis in SCA2”.

*2) Provide more details of experiments of Figure 3, regarding normalizing measurements between cells and genotypes, range of stimulus intensities, input-output analyses between genotypes (if performed).*

As detailed below, we now provide more detailed description of the analysis procedures for Figure 3. We have also analyzed stimulus intensities used in these experiments and find no significant difference between genotypes.

*3) Please comment on the observed reduction in Purkinje neuron phasic firing in SCA2 as early as 6-8 weeks, in contrast to the results here, which would predict enhanced firing frequency, ideally integrating with existing literature.*

Thank you for pointing out this paradox. In isolation, the increased mGluR1 coupling to TRPC3 cation channels would be expected to increase the excitability of PNs. However, because the increase in mGluR1-coupled cation conductance is also accompanied by changes in intracellular calcium levels, the effects on PN excitability are more complex. We have now added a paragraph to the end of the Discussion discussing how the increased mGluR function could result in the decreased spontaneous PN excitability that is characteristic of many SCA models.

*[…] Detailed points for essential revisions:*

*1) What the authors cannot conclude is the source of the raised calcium, but they make the inference that it is from IP3R dependent release from internal stores, based upon the literature and known signalling changes in SCA2. They speculate about this in the Discussion, and most likely there are multiple sources. Perhaps the authors should comment on the reduced calbindin expression in this SCA2 mouse, that would certainly exacerbate the feedback cycle that the authors propose.*

As indicated above, the revised manuscript now points out the loss of calbindin and its possible contribution to the elevated calcium.

*2) In the experiments shown in Figure 3, more details are needed about how these experiments were done and how they were normalized between cells and genotypes. What was the range of stimulus intensities used for PF stimulation? Did the authors perform an input-output analysis? If so, was there a difference between genotypes? How were calcium signals normalized between cells/genotypes?*

For the experiments in Figure 3, calcium imaging measurements were made in dendritic regions with local stimulation to activate PF excitatory synaptic inputs. PFs were stimulated with trains of different durations at 100 Hz (1 to 50 pulses). Each stimulus condition in each PN was repeated 5 times and these movies were averaged. The average responses were then background subtracted by defining a region of interest away from the cell (i.e., the background) and subtracting this value from a region of interest on the PN dendrite near the stimulating electrode. Background subtracted values versus time were then converted to ∆G/G values by determining the average baseline fluorescence value (Gbl) during the 1.5 seconds (32 frames) prior to stimulation; all time dependent values were than divided by Gbl and subtracted by 1 to yield ∆G/G for that PN. Mean values cross all WT and SCA2 PNs were then calculated by averaging these analyzed traces and are displayed for indicated conditions.

Due to practical limitations, we did not perform a full input output analysis nor did we systematically vary the stimulus intensity at individual sites. Rather we chose a stimulus intensity that gave a local calcium response; typically this stimulus intensity was 5-25 microamps / 100 microseconds.

As shown in the table below, stimulus intensities (in microamps) for WT and SCA2 responses were not significantly different at either the 12-16 week old or the 20-24 week old ages.

12-16 weeks20-24 weeksgenotypeWTSCA2WTSCA2Mean10.110.0317.617.14STD2.363.4911.88.9SEM0.470.784.173.37N232087

*3) The authors should comment on the observed reduction in Purkinje neuron phasic firing in SCA2 as early as 6-8 weeks, whereas in contrast the results here would predict enhanced firing frequency. Could the enhanced mGluR1 currents, and therefore Purkinje neuron excitability be particularly important during behaviourally relevant changes in Purkinje neuron firing behaviour eg. after high frequency parallel fibre input, in response to a sensory stimulation ? Or is tonic mGluR1 activity enhanced and firing frequency depressed through an alternative mechanism? (E.g. increase KCa channel current?)*

As mentioned above, the interaction of membrane excitability and intracellular calcium mechanisms will determine how changes in mGluR function impact spontaneous excitability. One working hypothesis involves SK-type, Ca activated K^+^ conductances. PNs have a large amount of SK K^+^ conductance and when these channels are inhibited (Womack and Khodakhah, 2003; Egorova et al., 2016) or activated (Walter et al., 2006; Maiz et al., 2012; Kasumu et al., Chem. Biol. 19:1340, 2012; Egorova et al., 2016) there are robust changes in PN firing frequency both in vitro (Womack & Khodakhah, 2003; Walter et al., 2006; Maiz et al., 2012; Kasumu et al., 2012) and in vivo(Egorova et al., 2016). We hypothesize that tonic activation of SK type K^+^ conductances through elevated basal calcium contributes to the slowing of spontaneous PN firing observed in slices from SCA2 mice (Meera, Pulst and Otis 2016). While this is a straightforward possibility, further experiments will be necessary to test this. Among the open questions are why positive SK modulators seem to selectively affect baseline firing and pattern of firing in SCA2 PNs (Kasumu; Egorova) and how the rate and pattern of PN firing is altered by disease in vivo (Egorova).